# Microsatellite Instability Analysis (MSA) for Bladder Cancer: Past History and Future Directions

**DOI:** 10.3390/ijms222312864

**Published:** 2021-11-28

**Authors:** Chulso Moon, Maxie Gordon, David Moon, Thomas Reynolds

**Affiliations:** 1Department of Otolaryngology-Head and Neck Surgery, The Johns Hopkins Medical Institution, Cancer Research Building II, 5M3, 1550 Orleans Street, Baltimore, MD 21205, USA; 2HJM Cancer Research Foundation Corporation, 10606 Candlewick Road, Lutherville, MD 21093, USA; maxiegordon@bellsouth.net (M.G.); david_moon@alumni.brown.edu (D.M.); 3BCD Innovations USA, 10606 Candlewick Road, Lutherville, MD 21093, USA; 4NEXT Bio-Research Services, LLC, 11601 Ironbridge Road, Suite 101, Chester, VA 23831, USA; treynolds@nextmolecular.com

**Keywords:** bladder cancer, microsatellite, molecular diagnostics

## Abstract

Microsatellite instability (MSI), the spontaneous loss or gain of nucleotides from repetitive DNA tracts, is a diagnostic phenotype for gastrointestinal, endometrial, colorectal, and bladder cancers; yet a landscape of instability events across a wider variety of cancer types is beginning to be discovered. The epigenetic inactivation of the MLH1 gene is often associated with sporadic MSI cancers. Recent next-generation sequencing (NGS)-based analyses have comprehensively characterized MSI-positive (MSI+) cancers, and several approaches to the detection of the MSI phenotype of tumors using NGS have been developed. Bladder cancer (here we refer to transitional carcinoma of the bladder) is a major cause of morbidity and mortality in the Western world. Cystoscopy, a gold standard for the detection of bladder cancer, is invasive and sometimes carries unwanted complications, while its cost is relatively high. Urine cytology is of limited value due to its low sensitivity, particularly to low-grade tumors. Therefore, over the last two decades, several new “molecular assays” for the diagnosis of urothelial cancer have been developed. Here, we provide an update on the development of a microsatellite instability assay (MSA) and the development of MSA associated with bladder cancers, focusing on findings obtained from urine analysis from bladder cancer patients as compared with individuals without bladder cancer. In our review, based on over 18 publications with approximately 900 sample cohorts, we provide the sensitivity (87% to 90%) and specificity (94% to 98%) of MSA. We also provide a comparative analysis between MSA and other assays, as well as discussing the details of four different FDA-approved assays. We conclude that MSA is a potentially powerful test for bladder cancer detection and may improve the quality of life of bladder cancer patients.

## 1. Introduction

Microsatellite instability (MSI) is a molecular tumor phenotype resulting from genomic hypermutability and is initially described as variations in the length of microsatellite sequences in the entire genomic structure. As part of the familiar colon cancer syndrome, MSI is most commonly observed among Lynch syndrome patients [1,2,3]. However, since its initial discovery, MSI has been acknowledged as a generalized phenomenon in a wide spectrum of sporadic cancers [4,5,6,7,8], and the underlying mechanisms for these cases of sporadic cancers seem to be based on epigenetic mechanisms, namely the methylation of MLH1. Probably the most deleterious outcome of MSI, in either inherited or sporadic sold tumors, is the frameshift mutations in tumor-associated genes, which can be accumulated over a long period of time. This process plays a crucial role in the various stages of human carcinogenesis. MSI detection is currently achieved by examining PCR products from a few important microsatellite markers (MSI–PCR) [9,10]. Recently, several groups have developed methods to analyze MSI using massively parallel DNA-sequencing technologies [11,12,13,14,15,16]. This new approach is not only based on a large number of samples but is also designed to offer robust quality and quantitative precision that was not previously achievable using the PCR technique. Notably, Cancer Genome Atlas (TCGA) made it possible to predict MSI status regardless of cancer through information from tumor exomes sequencing. This approach resulted in a more comprehensive understanding of genome-wide MSI [13]. Likewise, next-generation sequencing (NGS) also become a useful tool in examining MSI-positive carcinogenesis, in addition to discovering valuable biomarkers and novel therapeutic targets [15]. Here, we review the latest progresses in the investigation of MSI+ (MSI positive) carcinogenesis [17,18,19,20,21,22,23,24,25,26,27,28,29,30,31,32,33,34,35]. Bladder cancer (here we refer to transitional carcinoma of the bladder) has become one of the major causes of cancer-specific morbidity and mortality in various parts of the world [36]. Our understanding of its etiology, the molecular characteristics associated with its progression, and management guidelines render bladder cancer an ideal candidate for screening [36,37,38,39]. The diagnosis and follow-up of bladder cancer is still very difficult due to the lack of cancer-specific symptoms and therefore become a challenge for the medical community, not only for urologists but also for primary care physicians [36,39]. Cystoscopy, which has been used as a gold standard for the detection of bladder cancer, is naturally invasive, with frequent complications, while its cost is relatively high [39]. Urine cytology provides only limited diagnostic value due to its low sensitivity, particularly to low-grade tumors [36,39]. Due to urgent need, new qualitative and quantitative molecular tests designed to identify cellular and molecular changes exclusively associated with bladder cancer have been explored [38,39]. Therefore, over the last three decades, several new “molecular tests” for the diagnosis of urothelial cancer have been developed. In fact, studies of several molecular assays combined with traditional screening methods have demonstrated promising results [39]. LOH is typically identified by comparing the DNA isolated from tumors to normal DNA, such as that isolated from blood [7,8,9]. This LOH can be detected using a method known as microsatellite instability analysis (MSA).

Here, we review the discovery and evolution of MSI in cancer biology, as well as the development of different techniques for detecting MSI [4,6]. Next, we focus on the use of MSA for bladder cancer detection, focusing on its initial and later clinical development. We also discuss the use of MSA for the early detection of bladder cancer and its role as a tool in the surveillance of recurrent bladder cancer. Additionally, we discuss four different technical guidance approaches in the conclusion section.

The intention of this report is to provide a comprehensive narrative review and not a meta-analysis. We use the following review criterion: a literature review based on searching PubMed for original articles published between 1968 and 2021, using the search terms “bladder cancer”, “microsatellite instability” “urine biomarker”, “recurrent bladder cancer surveillance”, “LOH for bladder cancer detection”, and “bladder cancer management”. The search was limited to full-text articles written in the English language. Published reviews were used as additional sources of references.

## 2. An Overview of MSI

### 2.1. Initial Discoveries and Clinical Applications

Four pathways have been identified as causes of genomic instability for various cancer types, including gastrointestinal cancer [17,18,19,20]. First, chromosomal instability (CIN) phenotype, characterized by aneuploidy and multiple structural rearrangements in multiple loci of chromosomes, is found in the majority of solid tumors. Second, in initially discovered colon cancer, an accumulation of newly generated somatic mutations among oncogenes (K-ras) and tumor suppressor genes (APC and p53) can be a direct cause of genomic instability. Third, the accumulation of DNA demethylation can be a precursor to genomic instability among some subsets of gastrointestinal cancers [21,22]. Fourth, the MSI phenotype can result in multiple small insertions and deletions among repetitive sequences (microsatellites) and was reported in 10% to 15% of sporadic colon cancers. Therefore, various tumors associated with Lynch syndrome (HNPCC) and certain patients populations with gastrointestinal and endometrial cancer are associated with the MSI pathway [23,24]. Tumors with MSI phenotypes can carry accumulated somatic mutations, which are caused by alterations in multiple simple repeated sequences (microsatellites). 

In Figure 1, an example of CAG repeats (microsatellite) gains and losses due to DNA replication error and repair pathways correcting such mistakes is shown as a simple diagram [24]. A normal human genome contains numerous triple base repeats (such as the CAC repeats in Figure 1) and these repeats are prone to DNA replication errors due to DNA polymerase slippage, which can result in either the gain or loss of triplet repeat [23]. In normal cells, DNA mismatch repair (MMR) machinery guarantees t genomic fidelity by recognizing, with the help of the MSH2/MSH6 enzymatic complex, DNA replication errors; the MLH1/PMS2/1 complex then repairs these DNA replication mistakes. However, in MSI tumor cells, the presence of a deficient MMR (dMMR) system results in the failure to repair DNA mismatches in microsatellites, resulting in the accumulation of mutations in different genomic codons [23,24]. In summary, when the DNA mismatch repair system (MMR mechanism) does not faithfully execute its work, DNA replication begins to cause accumulated DNA strand misalignment, which then becomes permanent new genetic material, leading to permanent genomic instability. The fact that serious defects in the replication fidelity of these unstable sequences can result in ubiquitous somatic mutations is one of the key insights supporting “the mutator phenotype for human carcinogenesis” [17,18].

Based on initial studies on Lynch syndrome, multiple studies of MSI for various forms of human carcinogenesis have resulted in three key conclusions. First, impairments in the mismatch repair (MMR) system [23,24] can be caused by either the mutational deactivation of the enzymatic function or the DNA methylation-based silencing of several key genes in the MMR pathway [24]. Second, overall, MSI-positive (MSI+) cancers carry unique genetic, molecular, and clinical phenotypes, which can differentiate them from MSI-negative (MSI−) cancers. Third, MSI can cause multiple mutations, which can lead to the activation and deactivation of oncogenes and tumor suppressor genes, and a series of these processes then can become a pivotal force in driving human carcinogenesis.

Recent work suggests that MSI may be used as a predictor for immune-checkpoint-blockade therapy. Several clinical studies have shown better outcomes among patients with MSI-positive tumors as compared to negative groups when they are treated with inhibitors of programmed cell death 1 (PD-1). These observations can be explained by the ability of T lymphocyte to detect multiple peptides neoantigens produced by a variety of MSI-based DNA mutations [25,26]. While mutations resulting from MSI can also initiate multiple steps of oncogenesis, MSI signatures can be distinctive among different cancer types as different genomic loci can be unstable depending on the tumor type [11,28,29,30,31,32,33,34]. Moreover, depending on the tumor type, each MSI carries unique different prognostic meanings [29] with different frequencies [28]. Several parameters are used for MSI detection, and these parameters can change the accuracy for MSI detection among different cancer types as each cancer type differs according to the type of microsatellite markers. Several microsatellite panels have been proposed for the accurate detection of MSI, including the Bethesda/NCI panel, the gold standard microsatellite panel for MSI detection [28,35].

### 2.2. Evolution of MSA Methods

MSI detection methods have constantly changed through the pursuit of better accuracy and efficiency, which rely on the amplification of one or several microsatellite markers with PCR and the detection of MSI (Figure 2). MSI detection was first tested among colorectal cancer samples by using PCR on specific MSI markers with polyacrylamide gel electrophoresis and autoradiography [3]. Furthermore, a fingerprinting method based on arbitrarily primed PCR (AP-PCR) was combined with electrophoresis [2]. In some laboratories, MSI detection was performed with silver or ethidium bromide staining of polyacrylamide gel. These methods are laborious, costly, and time-consuming, often resulting in inconsistent accuracy. They have therefore been replaced by a newer approach, which is based on PCR with fluorescent primers and capillary electrophoresis using a DNA genetic analyzer. This new method has also been modified by the multiplexing of PCR to allow amplification of 2–5 microsatellite markers.

### 2.3. Loss of Heterozygosity (LOH) in Bladder Cancer Patients

Bladder cancer develops through several premalignant stages; the search for chromosomal markers that can serve as early neoplasia detection markers or predict recurrence has identified several genomic regions that contribute to neoplastic progression. Many studies have focused on the loss or inactivation of tumor suppressor genes (TSGs). TSGs can be inactivated by numerous mechanisms, including point mutation, loss of heterozygosity (LOH), homozygous deletion, and hypermethylation. LOH has emerged as a major marker of bladder cancer tumor progression [27,34]. TSGs are one of the most common genetic changes in human cancers and LOH can play a vital role in inactivating these genes. Abnormalities involving p16 (chromosome 9p21) and p53 (17p13) are associated with superficial transitional cell carcinoma (TCC) and these two genes are among the two most frequently observed areas of LOH in bladder cancer. LOH at 9p has been shown to offer prognostic value in non-muscle-invasive bladder cancer [36], as well as 18q, 4p, 16, 20, and 21 [36]. LOH is typically identified by comparing the DNA isolated from tumors to normal DNA, isolated from blood. 

## 3. Use of Microsatellite Assay for Bladder Cancer Detection

### 3.1. Bladder Cancer Overview and Diagnostic Biomarkers

The incidence and prevalence of transitional cell carcinoma of the bladder (‘urothelial carcinoma’), the most frequently observed type of urinary bladder cancer in the Western world, has significantly increased in the past two decades, while its disease control in terms of mortality rate has not improved [37]. Additionally, its management cost per patient in the US has become the highest among any cancers. In the USA alone, by 2015, the associated direct and indirect medical costs added up to $4 billion per year [38]. The absence of bladder tumor-specific symptoms makes both the initial diagnosis and the follow-up of bladder cancer a significant burden not only for urologists but also the medical community in general. Cystoscopy, the gold standard for the detection of bladder cancer, is invasive and sometimes carries unwanted procedures related complications, while its cost is relatively high. Most recently, new technologies such fluorescence or narrow-band imaging-assisted cystoscopy, have emerged as potential tools. However, their invasive nature and added costs still prevent them from becoming standard practice. Overall, any measures for improved accuracy, simpler, and less expensive diagnostic tests are urgently needed for the improved management of bladder cancer [39,40,41]. For a long time, the cytology test from voided urine has been a dependably specific, noninvasive supplementary test to cystoscopy. However, it features two major limitations. First, while it offers a reasonably acceptable sensitivity for identifying high-grade urothelial cancer, its rate of detection for low-grade tumors ranges from only 4% to 31% [42]. Second, the accuracy of cytology is largely based upon the subjective readings of pathologists, which makes it difficult to provide a dependable, high-quality reading. Naturally, due to the urgent need, new qualitative and quantitative molecular tests designed to identify cellular and subcellular alterations exclusively associated with bladder cancer have been explored. 

In fact, in the last two decades, several new “biochemical and molecular assays” for the diagnosis of urothelial cancer have been reported. What should we expect from these new tests? First, easier, better, faster, and cost-efficient ways of screening for bladder cancer have been regarded as the four gold standards for determining better methods for the surveillance of low-grade tumors with papillary features. Second, and at the same time, such noninvasive methods need to offer reasonably high sensitivity and specificity. Third, the goal of such tests needs to be aimed at decreasing the need for cystoscopies, thereby improving patient quality of life. Fourth, for high-grade disease, the improved sensitivity of markers can result in earlier identification of recurrent tumoral disease. As mentioned above, an ideal marker can be defined as a test that offers ‘easier, better, faster, cost-efficient’ cancer detection [43]. So far, several viable new urine markers for the diagnosis of bladder cancer have been reported. Many of them demonstrate superior sensitivity to that of standard urine cytology, and it is important to note that this improved diagnostic accuracy seems to be crucial in detecting low-to-moderate grade lesions, although none of them have been used as a standard diagnostic test in clinical management guidelines [44,45]. There are two different types of noninvasive bladder cancer: the frequently recurring papillary tumor (Ta), and the more aggressive carcinoma in situ (CIS). While either type can progress into invasive tumors (T1–T4), the development of low-grade Ta tumors into the invasive cancer is unlikely, while high-grade Ta tumors and CIS can result in invasive carcinoma. Noninvasive tumors have been grouped as ‘superficial bladder cancer’ and are different in their behaviors from invasive tumors with basement membrane invasions [46,47,48]. On average, 70–80% of bladder cancers are noninvasive and patients with noninvasive or organ-confined invasive tumors (T1/2, N−) carries receive a better prognosis than those with more advanced disease, including extravesical tumors (T3/4, N−), tumors with lymph node metastases (any T, N+), or metastases (any T/N, M+) [47,48,49].

It is important to note that despite improvements in management, such as the use of chemotherapy or chemo-radiation therapy, the overall prognosis of bladder cancer has not improved over the past three decades [37,38,39]. For example, as of 2010, the 5 year overall survival rate for bladder cancer patients was 79.8%, a 2.8% increase from 1974–1986. In summary, 92.5% of patients with bladder-only disease survive for 5 years, a 4.5% increase from 1974–1986. By contrast, the current 5 year survival rates for patients with more invasive disease with invasions to extravesical area/nodal area or distant metastasis are 44.7% and 6.1%, respectively, indicating a small decrease of 0.3% and 3.9%, respectively, since 1974–1986. Therefore, once bladder cancer progresses beyond organ-confined disease, the chance of successful management is minimal; early detection of bladder cancer seems to be the most effective way of securing a better prognosis.

In recent years, the focus on urine-based bladder tumor markers (UBBTMs) has been aimed at reducing the frequency of invasive follow-up for patients with a prior history of bladder cancer. Ideally, a rapid, inexpensive test with strong specificity and sensitivity would best serve the target population. However, the development of such ideal tests has not been achieved yet. As discussed briefly above, cytology has been hampered by its low sensitivity for low-grade tumors [49,50,51,52,53,54,55,56]. Among the assays evaluated in the past 20 years, several warrant further discussion. These assays include the following: the bladder tumor antigen (BTA) test, BTA TRAK, BTA stat, urinary nuclear matrix protein 22 (NMP22), fibrin degradation product (FDP), autocrine motility factor receptor, bladder cancer nuclear matrix protein (BCLA-4), cytokeratin 20, telomerase, hyaluronic acid, hyaluronidase, Immunocyt, the urinary bladder cancer (UBC) test, CYFRA 21-1, chemiluminescent hemoglobin, hemoglobin dipstick, urinary tissue polypeptide-specific (TPS) antigen, bladder cancer antigen (BCA), beta-human chorionic gonadotropin, tissue polypeptide antigen (TPA), and microsatellite analysis.

### 3.2. Overview: MSI and LOH in Bladder Cancer Detection

UCC is characterized by the frequent loss of heterozygosity (LOH) at the following chromosomal locations: 4p, 8p, 9pq, 11p, and 17p [36,57,58,59,60]. As discussed above, MSI targets tandem repeats in genomic DNA to evaluate the loss of heterozygosity (LOH) that occurs with tumor cell transformation [60,61,62,63,64,65,66,67]. These MSI biomarkers, originally developed at Johns Hopkins University, have been shown to detect deletions in DNA isolated from the urine sediment of bladder cancer patients before the cytoscopic detection of tumor [68,69]. Several studies have shown that these microsatellite changes can be profiled in urine for the detection of bladder cancer cells [70,71,72,73,74,75,76]. Microsatellite analysis for bladder cancer detection is based on 15 to 20 markers from a region with a high percentage of LOH during the development of bladder cancer. This method proved very sensitive for low- and high-grade lesions with sensitivities of 67%, 86%, and 93% for recurrent G1, G2, and G3 lesions, respectively, and an initial specificity of 88% [68,69,70,71,72,73,74,75,76,77,78,79,80]. Importantly, microsatellite analysis could predict a recurrence before cystoscopically detected evidence in all studies with extended follow-up [79]. Although microsatellite analysis (MSA) offers extremely high potential, automation of the assay, multi-center studies, and data interpretation for patients with persistent leukocyturia are crucial to make it a widely used test. 

### 3.3. Microsatellite Analysis: Initial Studies

As discussed above, studies on the applicability of MSI analysis to the diagnosis of bladder tumors were first described by Mao et al. at Johns Hopkins University [69]. In this study and other studies cited below, the MSI analysis was considered positive for tumor if any one of the 15 microsatellite markers was positive in their initial report. These investigators described the analysis of urine and tumor specimens obtained from groups of individuals who had received transurethral resection of known bladder tumors (TURBT). Control specimens and material obtained from patients treated for non-malignant bladder conditions were also analyzed. After the extraction of genomic DNA from the peripheral lymphocytes of each study participant, DNA was also extracted from urine specimens obtained prior to tumor resection. DNA was also retrieved from each tumor specimen. Using a panel of microsatellite markers previously shown to be highly informative in bladder tumors, electrophoresis of the PCR products revealed a high rate of concordance between microsatellite alterations in tumors and urine specimens from the same patients. In this small initial experience, the sensitivity of MSI analysis was estimated to be 95% for the detection of various clinical stages of bladder cancer [68,69].

In a subsequent study, Steiner et al. used MSI analysis (MSA) as a monitoring tool for the surveillance of patients after treatment. Utilizing the same technology, a group of 21 patients was followed with cystoscopy, cytology, and microsatellite analysis. The investigators reported that microsatellite analysis anticipated the development of bladder tumors as long as 6 months prior to cystoscopy [77]. The rate of cancer detection exceeded 90% (10 of 11 cancers detected by MSA). The investigators also reported a specificity of 100% [80]. An additional study reported that the sensitivity and specificity of MSA on frozen archived urine samples were both over 80%, which was comparable with other tests on the market [81]. Other groups have examined the diagnostic utility of microsatellite analysis within the context of bladder cancer management. Mourah et al. (1998) studied 59 patients with bladder cancer and reported an MSA sensitivity of 83% and a specificity of 100% [82]. Similarly, Steiner et al. (1999) reported a diagnostic sensitivity for their 20 marker MSA assay of 91% [83], and Baron et al. (2000) found MSA sensitivity to be 84% in their patients with bladder cancer. Similar findings were reported among bladder cancer patients with cystitis [84,85]. MSA results may occasionally be confounded by intercurrent inflammatory processes, although this appears to be a rare event and requires study in a large group of patients [85,86]. MSA sensitivity appears to increase with stage and grade of disease in the small number of patients evaluated to date [86,87,88]. These studies also indicated that fewer than 20 markers can be sufficient to provide high sensitivity and specificity. In addition, the microsatellite markers employed in the MSA assay must be tailored to the population under scrutiny, as demonstrated by the initially low informativeness of the assay in Chinese patients studied using marker panels that were developed for occidental patient cohorts [87,88,89]. These studies suggest that as experience with MSA increases and as marker panels expand and become better defined, MSA tests will achieve high sensitivity and specificity, making them an ideal approach for a large clinical trial and, ultimately, for evaluation as a substitute for standard-of-care technology. In summary, based on 9 studies that tested the diagnostic accuracy of MSA as an initial diagnostic tool (Table 1A), sensitivity was 90%, which resulted from the detection of 373 out of 417 cancer samples. Specificity was 98%, which resulted from the detection of 113 out of 115 normal samples. Notably, most of these studies were performed in the early 2000s and the most recent study was performed in 2009. No meaningful report has emerged in the last 10 years.

### 3.4. Microsatellite Analysis: Follow Up Studies

Bartoli et al. reported that out of a total of 120 patients who visited a urological office, including 73 individuals with transitional cell carcinoma and 43 individuals who served as controls [90,91], MSA was performed in the blood/urine pair using 3 multiplex polymerase chain reactions per patient, covering 10 MSI markers. Notably, urine sediment inflammatory cells were assessed through a urine dipstick test and an ABI Prism^®^ 310 Genetic Analyzer was used to calculate the cutoff for allelic imbalance. Out of a total of 66 patients who had microsatellite analysis alterations in their urine sediment, 59 had transitional cell carcinoma, while the test’s sensitivity and specificity were 80.8% and 85.1%, respectively. Notably, statistical analysis did not suggest any noticeable influence of inflammatory status by urine dipstick test on MSA performance. This study also confirmed that chromosome 9 plays a specific role in transitional cell carcinoma, since among the control samples, the allelic loss of chromosome 9 was seldom seen. This multiplex microsatellite analysis method carried a low cost and took less time than prior methods, while this study was based on 10 different markers.

Another study by Bartoli et al. showed similar findings [90,91] to their first report, while in this report, 13 MSI markers were used instead of 10. Among 42 patients, the diagnostic accuracy of four different tests, including urinary cytology, urinary bladder cancer (UBC) marker, bladder tumor antigen (BTA), and MSA were compared. Monoclonal antibody tests were used for the UBC and BTA-t analyses, while a urinary Autocyte Preparation System^®^ was used for the urine cytology. First, urinary cytology detected a lower number of bladder cancer cases than expected: 13.3% of tested TCC (bladder cancer) patients. Second, while 73.3% of cancer patients were detected by BTA-t marker with 50% specificity, the UBC marker detected 63.3% of cancer patients with a 41.6% specificity [90]. Third, MSA identified 83.3% of cancer patients with a 100% specificity. Fourth, and most importantly, MSA demonstrated high sensitivity among patients with superficial (81.4%) disease or G1 (80%) grade tumors, where cytological studies detected little or no cancer cases. This study concluded that 13 marker MSA is a highly sensitive and specific test for the initial diagnosis of TCC and recurrence monitoring, and it offers powerful diagnostic accuracy among patients with superficial, early-stage, and low-grade tumors. Frigerio et al. [72] found that cytology combined with MSA offered high sensitivity for identifying primary tumors and could detect all recurrent diseases. In a prospective study among 91 patients, LOH analysis with MSA, UroVysion FISH, and conventional urine cytology were compared with the histological findings of consecutive transurethral biopsies. Although all the samples could be analyzed by our LOH assay, only 56 samples were suitable for all 3 analyses. The highest sensitivity was obtained with our LOH-assay/cytology approach (G1-2: 72%; G3: 96%), which was only surpassed by a combination of all three techniques (G1-2: 83%; G3: 100%). Importantly, over 93% of the patients with recurrent disease could be detected by LOH/cytology analyses of their voided urine samples; a surveillance plan based on alternating cystoscopy with LOH/cytology-examination was proposed. Likewise, van Rhijn et al. [92,93,94] demonstrated that out of 47 Caucasian patients with confirmed superficial bladder TCC (37 pTa, 10 pT1) at initial diagnosis, MSA demonstrated a 94% (44/47) sensitivity. Overall, the studies by Bartoli, Frigerio, and van Rhijin firmly confirmed that microsatellite assay is an efficient way of detecting early stage bladder cancer.

Wild et al. reported, among 119 patients [76], diagnostic accuracy for loss of heterozygosity (LOH), FGFR3 mutation, polysomy, and p16 deletion using UroVysion FISH. From voided urine samples, initially, the three methods (CYTO, LOH, and FGFR3) were assessed individually for their diagnostic accuracy to detect bladder cancer cells in urine. Additionally, FGFR3 and LOH analysis were tested in conjunction with cytology. The combination of the three techniques (CYTO + LOH + FGFR3) did not provide significant diagnostic accuracy. However, the combination of cytology with LOH analysis (CYTO + LOH) resulted in the highest diagnostic accuracy for the detection of bladder cancer cells and performed better than cytology alone (sensitivity of 88.2% and specificity of 97.1%). The study further concluded that a combination of cytology with LOH analysis may be able to reduce the number of unpleasant cystoscopies for bladder cancer patients.

In summary, based on 9 studies that tested the diagnostic accuracy of MSA in a recurrent surveillance setting (Table 1B), sensitivity was 87%, which resulted from the detection of 251 out of 290 cancer samples. Specificity was 94%, which resulted from the detection of 185 out of 196 normal samples. Notably, most of these studies were performed in the early 2000s. No meaningful report has emerged the last 10 years. Besides, some of the studies {76} reported data in combination with other molecular assays and are therefore not included in Table 1B.

### 3.5. MSA Assay for Surveillance for Recurrent Bladder Cancer

Several independent reports have previously suggested the higher sensitivity of MSA (75–96%) compared to that of cytology (13–50%) in various settings of clinical testing designs [69,70,71,72,73,74,75,76,77,78,79,80,81,82,83,84,85,86,87,88,89,90]. Unlike conventional cytology, MSA detected low-grade and low-stage disease as efficiently as it could high-grade and high-stage disease [81,82,83,84,85,86,87,88,89,90,91,92,93,94,95,96,97,98]. Amira et al. were one of the first groups who analyzed MSA results in a more comprehensive study design [79] and reported that a positive MSA test preceded a visible recurrence between 1 month and 9 months in 75% of patients in their follow up cohorts. Similarly, a report by van Riijen [99] demonstrated that, out 93 bladder cancer patients, MSA identified 18 of the 24 recurrent tumors, while the six undetected tumors turned out to be minimal pTaG1 lesions, for which early surgical treatment was not necessary. By contrast, 5 out of 9 patients with a positive MSA with a negative cystoscopy, as in the report by Amira et al., had a tumor recurrence within 6 months that was detected by cystoscope. Overall, the sensitivity (74%) was significantly higher than that of the BTA stat test (56%) or urine cytology (22%). 

As discussed above, Frigerio et al. [72] reported that the combined use of cytology and LOH analysis could detect most diseases. They obtained a sensitivity of 72% for grade 1–2 tumors and 96% for grade 3 tumors. Likewise, van Rhijn et al. [78] demonstrated that out of 47 patients with superficial bladder TCC (37 pTa, 10 pT1) at initial diagnosis, who had been followed up for 12–48 months after tumor removal, MSA provided a precise detection of cancer: 94% (44/47) of primary tumors and 92% (12/13) of tumor recurrences. As discussed above for two previous study reports, a majority (5% (9/12)) of tumor recurrences were discovered 1 to 9 months prior to cystoscopy-based detection of recurrent disease. This study’s results are consistent with the notion that MSA can reliably detect superficial bladder tumors at the time of diagnosis, while it can also be used for the detection and prediction of recurrent tumors. 

Notably, van Rhijn et al. performed an extensive literature review on the use of 18 markers (BTAstat, BTAtrak, NMP22, FDP, ImmunoCyt, Cytometry, Quanticyt, Hb-dipstick, LewisX, FISH, Telomerase, MSA, CYFRA21-1, UBC, Cytokeratin20, BTA, TPS, and Cytology) in the surveillance of recurrent bladder cancer [94]. The highest median sensitivities were reported for CYFRA21-1 (85%), Cytokeratin20 (85%), and MSA (82%). The highest specificities were reported for Cytology (94%), BTA (92%), and MSA (89%) (Table 2 and Table 3). This report concluded that MSA, ImmunoCyt, NMP22, CYFRA21-1, LewisX, and FISH are the most promising markers for surveillance at the time of writing. Nevertheless, so far, there is not enough clinical evidence to warrant the substitution of the cystoscopic follow-up scheme by any of the currently available urine marker tests. Likewise, currently, the data are not consistent with the sole use of molecular tests in patients with a high risk of developing bladder cancer. However, many studies have shown molecular tests to offer value in not only improving the diagnostic accuracy of high-risk groups in the initial diagnosis of bladder cancer, but also in the prediction of recurrence [91,92,93,94,95,96,97,98,99,100,101,102], albeit only when used in conjunction with cytology and cystoscopy; molecular testing can reduce the need for these procedures. 

### 3.6. MSA Assay as a Tool Predicting Recurrent Bladder Cancer

Amira et al.’s report was one of the first to examine MSA results after a systemic surveillance design [79]. This study group found that a positive MSA test preceded a visible recurrent disease by 1 month to 9 months among 75% of patients under surveillance. Similarly, the report by van Riijen demonstrated that out of 93 bladder cancer patients in a surveillance setting, 5 of 9 patients with a positive MSA with a negative cystoscopy had a tumor recurrence within 6 months after urine collection [93,94]. Importantly, van der et al. reported that out of 228 patients monitored by MSA as a prospective surveillance tool, the two-year risk of developing recurrent disease reached 83% if the MSA outcome was persistently positive, while it was 22% when MSA was persistently negative [75]. Overall, the author suggested that MSA status can be used as a potentially powerful tool in predicting recurrent disease. In subset analysis, the positive predictive value of MSA for recurrent disease is much higher in patient groups with an FGFR3 wild-type from resected tumor samples, as it reaches 100% at 24 months of follow-up [96,97,98]. This finding is consistent with the observation that FGFR3 gene mutations are known to be associated with genomic stability in bladder cancer [97,98,99,100,101,102,103]. Notably, several studies have identified that MSA more frequently fails to detect recurrent disease among nonsmoking patients. Moreover, surveillance by MSA seems to be more efficient in the assessment of smoking patients with an FGFR3 wild-type tumor, which is genetically more unstable [96,97]. For patients with an FGFR3 mutant tumor, surveillance by FGFR3 mutation analysis was proposed as a potentially viable choice [96,97,98].

### 3.7. MSA Assay for Different Ethnic Group

Notably, the microsatellite markers employed in the MSA assay must be tailored to the population under scrutiny, as demonstrated by the initially low informativeness of the assay on Chinese patients studied using marker panels that were developed for occidental patient cohorts [104,105,106,107,108,109,110,111]. These studies suggest that as experience with MSA increases and as marker panels expand and become better defined, MSA tests will achieve high sensitivity and specificity, making them an ideal approach for a large clinical trial and, ultimately, for evaluation as a substitute for standard-of-care technology. Recent studies performed in China on STR markers have established that the heterozygosity frequencies for many of the STR markers is sufficient for LOH testing in cancer. In their report, Song et al. investigated the application of 13 short tandem repeat (STR) loci (D13S317, D7S820, TH01, D16S539, CSFIPO, VWA, D8S1179, TPOX, FGA, D3S1358, D21S11, D18S51, and D5S818) routinely used in forensic analysis for delineating population relationships among seven human populations representing the two major geographic groups, namely the southern and northern Chinese [104]. The resulting single topology suggested significant geographic and population partitioning, which was in line with the differences in geographic location, languages, and eating habits. These findings also suggest that forensic STR loci can potentially become powerful tools as they can provide the necessary fine resolution for reconstructing recent human evolutionary history. Another study tested the genetic polymorphism of 29 STR loci in the Hunan, Han population in China, and further investigated the application of short tandem repeat (STR) loci routinely used in forensic analysis in Chinese populations [111]. Both studies revealed heterozygosity frequency (HF) for FGA and TH01, two of the MSA markers most frequently used in the Chinese population. HF needs to be above 50% to be considered a good marker for MSA and TH01 and FGA showed 86% and 66% HF, respectively. The correlation of HF with STR markers in this and other studies was >50% HF in general in STR markers in the Chinese population and, therefore, to extrapolate that expected HF in the MSA markers. We recently proposed a multiplex PCR format and proposed that 15 markers (manuscript submitted) would achieve at least five informative markers in the Chinese population in 99.995% of the patients if the HF frequency is >50% per marker. 

### 3.8. Automated MSA Assay for Detection of Bladder Cancer

LOH is typically identified by comparing the DNA isolated from tumors to normal DNA, such as that isolated from blood. Short tandem repeat (STR) regions, also known as microsatellite regions, within chromosomes are unstable in cancerous cells and are deleted, causing a loss of heterozygosity (LOH) in tumor samples. This LOH is detected using a method known as microsatellite analysis (MSA). Traditionally, MSA is performed through the amplification of STR loci, using polymerase chain reaction (PCR) with primers flanking the STR region followed by polyacrylamide gel or capillary electrophoresis. STRs are short sequences of DNA, normally 2–5 base pairs in length, that are repeated numerous times in a head–tail manner, i.e., the 16 bp sequence of “gatagatagatagata” would represent four head–tail copies of the tetramer “gata”. The number of repeats in STRs varies not only from person to person but from one allele to another within the same person. Therefore, a person may possess six sets of a tetramer repeat on one allele and ten on the other allele; this person is said to be heterozygous at this STR region. PCR amplification of a heterozygous STR yields PCR products of two different sizes. Furthermore, these STR regions become unstable during cancer progression and may be lost due to deletion. When this loss occurs, only one PCR product is amplified in the tumor. Comparison of the normal DNA (isolated from peripheral blood) and the tumor specimen shows a loss of heterozygosity (LOH) in the tumor specimen (Figure 2). In this manner, a comparison of the normal DNA with the tumor DNA can detect the genetic changes indicative of cancerous lesions. 

There are three separate steps involved. First, polymerase chain reaction (PCR) amplification of the STR markers of matched blood and urine sediment genomic DNA: PCR is accomplished through PCR amplification using primer sets that flank the target STRs at 15 microsatellite loci. The 5′ end of each primer pair is fluorescently labeled to allow for detection of the PCR fragments by capillary electrophoresis. Second, in fluorescence-based fragment detection of the amplicons, the PCR amplicons are resolved on a capillary-based gel electrophoresis system that detects, sizes, and determines the relative fluorescence units (RFU) for each fragment. Third, in the determination of microsatellite instability status of each sample, the RFUs of heterozygous alleles detected in the blood are compared to the RFUs detected in the matched urine sample, and the ratio of RFUs from urine alleles to blood alleles is calculated. Markers that exhibit values outside the ratios seen in normal samples are said to exhibit a loss of heterozygosity (LOH). This LOH serves as an indicator of bladder cancer. 

A new approach that combines a PCR with fluorescent primers and capillary electrophoresis, which is performed by automatic DNA sequencer allowing fragment analysis at single base resolution has been pursued. This approach has also been modified by the multiplexing of PCR to allow the amplification of 2–5 microsatellite markers and the automatic identification of the allele size [112,113,114,115,116,117,118,119,120]. Clearly, this approach has been applied into MSA for bladder cancer detection. Barlott et al. [91,92] produced the first report using multiplex PCR for MSA. This group evaluated the feasibility of MSA using MPX PCR with fluorescent markers as a way of developing a simple, inexpensive method of high-sensitivity urine sediment diagnosis of TCC. Three sets of PCR tube were used in their study: Chromosome Analyzed Locus Analyzed MPX PCR 1: 9p23–p22 D9S162 9p22 IFN- 9p21 D9S171 9p21-p22 D9S747 MPX, PCR 2: 4q2.8 FGA 4pter-qter D4S243 16pter-qter D16S310 18q21.33 D18S51, and Single PCR: 18q23-qter MBP-LW 18 MBP-H. In this study, Barlott et al. [92] reported the test sensitivity and specificity to be 80.8% and 85.1%, respectively. The study suggested that multiplex microsatellite analysis can be a noninvasive, rapid, inexpensive, and reproducible method for screening for and monitoring superficial transitional cell carcinoma.

Most current microsatellite analyses are designed to compare the ratios of amplification products of the paternal and the maternal allele from the urine samples of patients against the ratios from their blood leukocytes as normal control, judging values below a range of 0.5–0.7 and over 1.5–2, respectively, as indications of a LOH/allelic imbalance. These thresholds were arbitrarily set and consequently did not take into account technical influences, such as differences in DNA quality and quantity between control and test samples or locus-specific amplification reproducibility, and could result in inconsistent findings in different groups. The thresholds also caused decreased sensitivity and sensitivity. Therefore, not only a good selection of informative markers but also the right ratio is crucial in developing a dependable MSA for bladder cancer detection. Commonly applied parameters are suggested based on rather conservative estimates to avoid false positives and thereby to secure a relatively high specificity, resulting in reduced sensitivity (minimal proportion of 35–50% tumor cells required).

To address this problem more systematically, Frigorie et al. experimentally determined individual threshold values for 10 STR-markers commonly affected in urothelial tumors by first comparing a large set of DNAs from normal human control tissues among themselves [72]. The threshold values from a retrospective study on biopsies from confirmed bladder carcinoma patients were tested in a blinded prospective survey for the sensitivity and specificity of MSA and these data were further compared with conventional cytological and UroVysion FISH analyses that were performed in parallel. PCR was performed for each patient. The authors proposed the 10 most informative markers. These included 5 marker loci on chromosome 9 (IFNA, D9S925, D9S162, D9S938, D9S747) previously linked to early tumor development and a set of markers on chromosome 17p (TP53, D17S960), 8p (D8S261), 13q (D13S155), and 16q (D16S476) [121,122,123,124]. The report also included a new threshold for each marker; ratios below 0.8 were linked to an LOH for the highly reproducible markers IFNA, D9S747, D16S476, and D17S960, while LOH thresholds between 0.5 and 0.7 were obtained for the technically less reliable marker amplifications of the microsatellite loci D8S261, D9S162, D9S925, D13S155, and TP53.

## 4. Discussion

Its multifaceted clinical presentation and expected course of progression make bladder cancer a potentially valuable screening target; currently, the consensus is that high-risk populations should be screened [39,40]. Two key aspects underscore the importance of bladder cancer screening in the coming decades. First, the persistently high prevalence of smoking is expected to function as key hazard for significant carcinogenic effects on the bladder for the next several generations. Second, it is highly unlikely for bladder cancer to have the capacity to metastasize before it becomes invasive [47]. Therefore, there is a valuable opportunity to detect bladder cancer early, in a window of timing between tumor origination and invasion. Certainly, the management of noninvasive cancers involves fewer morbidities and is more effective for curative treatment than that of invasive bladder cancer [51] as, in this stage of tumor development, cystectomy, systemic chemotherapy or chemo-radiation therapy are not required.

MSA for the detection of cancer poses several significant technical challenges, specifically in the area of allele calling and interpretation. The results obtained mirror the results obtained by Butler et al. during validation studies for STR analysis for human identity. Laboratory-to-laboratory variability due to instrumentation and personnel created differences in assay performance. Additionally, both stochastic effects and variations in peak height effects between the DNA derived from urine sediment samples and the DNA derived from blood samples provided more variation in the results, which added to interpretation differences. These differences were especially evident in samples that produced results slightly above or below the cut-off ratios established for LOH. Hence, STR assays present significant challenges, including establishing the parameters to be used for determining LOH from potential tumor cells isolated from urine sediment.

LOH is commonly observed among many different types of solid tumors and allows the detection of recessive loss-of-function mutations in tumor suppressor genes [57,58,59,60]. The detection of recurrent LOH in a genomic region is critical evidence for the localization of tumor suppressor genes. Multiple factors play a role in interpretation because most clinical samples collected from urine contain a mixture of tumor and normal cells, producing a potential mixture at each loci being analyzed, which can obscure losses of genetic material in tumor cells. Moreover, LOH can only be determined by comparative analysis of a control profile obtained from a blood sample. This study illustrates the challenge of qualifying a technically difficult biomarker assay that requires interpretation by an analyst. Recently, we successfully reproduced a dependable 15 marker, 3 multiplex PCR assay. Additionally, we are preparing a manuscript for a concordance study between three laboratories and two different genetic analyzers. Moreover, we are also preparing a report on a unique approach to MSA involving the use of genomic DNA from a buccal swab instead of blood samples. This new approach will be likely to be patient-friendly as this will clearly avoid unnecessary blood sampling.

As discussed extensively above, many independent groups previously confirmed the superior sensitivity of MSA (75–96%) compared to cytology (13–50%) in various clinical settings [57,58,59,60,61,62,63,64,65,66,67,68,69,70,71,72,73,74,75,76,77,78,79,80,81,82,83,84,85,86,87,88,89,90,91,92,93,94,95,96,97,98,99,100,101,102] and, based on 18 different studies from 1997 to 2009, the sensitivity was approximately 90 % and the specificity was close to 95% to 100% percent, depending on the clinical setting (Table 1). Unlike conventional cytology, it appears that microsatellite analysis, MSA, can detect low-grade and low-stage disease as accurately as high-grade and high-stage disease [68,69,70,71,72,73,74,75,76,77,78,79,80,81]. As discussed above, Frigerio et al. [72] reported that the combined use of cytology and LOH analysis by MSA carried high sensitivity for identifying primary tumors as well as for detecting almost all recurrent diseases. Likewise, van Rhijn et al. [92,93,94] analyzed 47 Caucasian patients with confirmed superficial bladder TCC (37 pTa, 10 pT1) at initial diagnosis and proposed three important observations on MSA. First, MSA correctly identified 94% (44/47) of primary tumors. Second, MSA correctly identified 92% (12/13) of tumor recurrences. Third, MSA predicted the chance of future recurrences 75% of the time (9/12); the tumor recurrences were molecularly detected 1–9 months before cystoscopy evidence of recurrent disease.

Several factors are involved in maximizing the sensitivity and specificity of MSA. First, by establishing a robust genetic marker profile and determining for each of the analyzed microsatellites individual marker threshold values for LOH/allelic imbalance, the sensitivity of tumor DNA detection can be achieved without compromising its specificity. Second, a careful set-up of dependable methods is necessary to avoid erroneous LOH- judgements due to PCR artefacts, repeatedly described by others. Third, performing MSA on a genetic analyzer, which is a commonly used tool, needs to be standard in MSA practice, since it offers two major advantages. First, sample processing can be largely automatized, and results are provided in the form of a numerical data readout, independently of inter-observer variability associated with the complex interpretation of morphological features. Moreover, the determination of LOH ratios on this platform is highly reproducible and provides reliable results even in situations when the cell conservation is suboptimal for cytological evaluation and/or FISH. In a way, the use of a standard genetic analyzer such as the ABI 3500 machine provides MSA with a significant advantage as MSA is affected little by changes in PCR conditions or the amount of genomic DNA applied. Third, the amount of genomic DNA to be analyzed needs to be sufficient; we suggest the use of least 20 to 30 ng of urine genomic DNA, 20 ng for 10 markers, and 30 ng for 15 markers. Fourth, we propose that at least 15 mL of urine should result in more than 855 of samples with enough urine DNA. In summary, while the field of MSA-based bladder cancer detection has been quite inactive in the last 8 to 9 years, it is still a viable option as a powerful biomarker for bladder cancer detection. However, as mentioned above, four key technical considerations need to be carefully addressed in order for this to be the case.

In summary, in this review, based on over 18 publications with approximately 900 sample cohorts, we provided the sensitivity (87% to 90%) and specificity (94% to 98%) of MSA. We also provided a comparative analysis between MSA and other assays, as well while as discussing the details of four different FDA-approved assays. We suggest that MSA can be a potentially powerful test for bladder cancer detection and may improve the quality of life of bladder cancer patients.

## Figures and Tables

**Figure 1 ijms-22-12864-f001:**
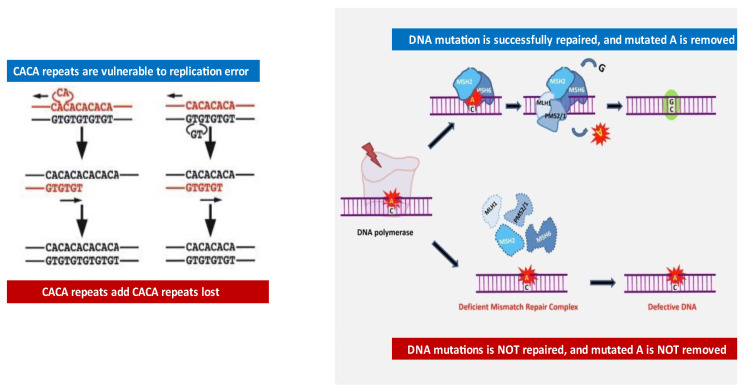
Development of repeat base abnormalities and role of mismatch repair in maintaining genomic fidelity. In the left side of the figure, an example of CAG repeat (microsatellite) gains and losses during DNA replication is shown. In the right side of the figure, the role of the mismatch repair complex in preventing replication errors is described. In normal cells, the DNA mismatch repair (MMR) machinery guarantees genomic fidelity by recognizing (via MSH2/MSH6 complex) and repairing (via MLH1/PMS2/1 complex) genetic mismatches generated during DNA replication. When normal G/C base pairs are mutated into A/C base pairs during DNA replication, the repair system recognizes the error (through MSH2/MSH6) and mutated A is then removed and replaced by correct C base via MLH1 and PMS1/2 machinery. Conversely, in MSI tumor cells, the presence of a deficient MMR (dMMR) system results in the failure to repair DNA mismatches in microsatellites, resulting in the accumulation of mutations in different genomic codons. So far, MLH1, MSH2, MSH6, and PMS2/1 have been found to be the main components of the MMR machinery. Modified from figure by Puliga E et al. [24].

**Figure 2 ijms-22-12864-f002:**
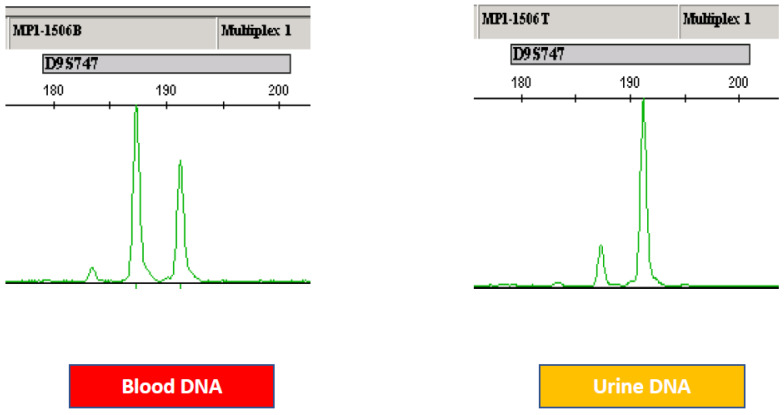
Example of MSA by genetic analyzer. In the upper panel, an MSI marker, M1506, is shown. In the left panel, a germ line MSI pattern is shown with two major peaks. In the right panel, both of the two peaks have been changed. These changes are detected by a series of algorithms in the ABI genetic analyzer.

**Table 1 ijms-22-12864-t001:** **A**: List of prior publications and sensitivity/specificity analysis on MSA study for the initial detection of bladder cancer. **B**: List of prior publications and sensitivity/specificity analysis on MSA study for the recurrent disease detection (surveillance setting) of bladder cancer.

A				
Study	No. of Cancers Detected	Sensitivity	Healthy Controls with Neg MSA Result	Specificity
	by MSA	(%)		(%)
**Mao et al. (1996) ♦ Science 271:659–662**	19/20	95	5 out of 5	100
Wild et al. (2009) ♦ Cancer Epidemiol. Biomark. Prev, 18, 1798–1806	71/81	88	37 out of 38	97
Linn et al. (1997) ♦ Int J Cancer 74:625–629	13/15	87	N/A	N/A
Schneider et al. (2000) ♦ Cancer Res 60:4617–4622	87/103	84	N/A	N/A
Sourvinos et al. (2001) ♦ J Urol 165:249–252	26/28	93	10 out of 10	100
Zhang et al. (2001) ♦ Cancer Lett. 172:55–58	73/81	90	19/19	100
Seripa et al. (2001) ♦ Int J Cancer 95:364–369	33/34	97	11 out of 11	100
Zhang et al. (2001) ♦ JNCI 93:45–50	22/23	96	17/17	100
van Rhijn et al. (2003) ♦ Clin. Cancer Res. 9, 257–263	29/32	91	14 out of 15	93
**OVERALL**	**373/417**	**90%**	**113/115**	**98%**
**B**				
**Study**	**No. of Cancers Detected**	**Sensitivity**	**Healthy Controls with Neg MSA Result**	**Specificity**
	**by MSA**	**(%)**		**(%)**
**van Rhijn et al. (2001) ♦ Cancer 92:768–775.e 271:659–662**	23/29	79	66 out of 70	94
Steiner et al. (1997) ♦ Nat Med. 3:621–624	10/11	91	10 out of 10	100
Baron et al. (2000) ♦ Adv Clin Path.4 (1):19–24	21/25	84	N/A	N/A
Bartoletti et al. (2005) ♦ Oncol Rep;13:531–537	25/30	84	30 out of 30	100
Bartoletti et al. (2006) ♦ J Urol175:2032–2037	59/73	81	36 out of 43	84
Bas et al. (2003) ♦ European Urology 43, 369–373		83		93
Frigerio et al. (2007) ♦ Int. J. Cancer Res, 121, 329–338	59/63	93	28 out of 28	100
Mourah et al. (1998) ♦ Int. J. Cancer Res. 79, 629–633.	10/12	96	15 out of 15	100
Amira et al. (2002) ♦ Int J Cancer 101:293–297	44/47	94	N/A	N/A
**OVERALL**	**251/290**	**87%**	**185/196**	**94%**

**Table 2 ijms-22-12864-t002:** Sensitivity and specificity of the data for various urinary biomarkers for surveillance of recurrent bladder cancer. Adapted from van Rhijn et al. [94].

Marker	Median Sensitivity	Range (Min–Max)	5% Difference	Median Specificity	Range (Min–Max)	5% Difference
BTAstat	70	24–89	Yes	75	52–93	–
BTAtrak	69	57–79	–	65	48–95	–
NMP22	73	47–100	–	80	56–95	Yes
FDP	61	52–81	Yes	79	75–96	Yes
ImmunoCyt	83	50–100	Yes	80	69–90	Yes
Cytometry	60	45–83	–	80	36–87	–
Quanticyt	59	45–69	–	79	70–93	–
Hb-dipstick	52	41–95	Yes	82	68–93	–
LewisX	83	80–89	Yes	85	80–86	–
FISH	84	73–92	Yes	95	92–100	Yes
Telomerase	75	7–100	Yes	86	24–93	na
Microsatellite	91	83–95	Yes	94	89–100	Yes
CYFRA21-1	94	74–99	Yes	86	67–100	–
UBC	78	66–87	Yes	91	80–97	–
Cytokeratin20	91	82–96	Yes	84	67–97	Yes
BTA	50	28–80	–	86	66–95	–
TPS	72	64–88	Yes	78	55–95	–
Cytology	48	31–100	Yes	94	62–100	–

**Table 3 ijms-22-12864-t003:** The median sensitivity per grade (G1–3) and specificity of the urinary biomarkers for surveillance of recurrent bladder cancer. Adapted from van Rhijn et al. [94].

Marker (Reference Number)	No. pts./Median Sensitivity			No. pts./Median Specificity
	G1	G2	G3	
BTAstat	228/45	206/60	208/75	972/79
BTAtrak	60/55	61/59	101/74	195/66
NMP22	56/41	77/53	81/80	235/59
FDP	13/62	36/64	22/86	113/80
ImmunoCyt	23/78	10/90	18/100	83/62
Cytometry	18/11	54/41	38/66	52/87
Quanticyt	-	11/64	5/80	56/68
Hb-dipstick	13/15	36/39	22/73	113/87
FISH	25/56	9/78	20/95	130/70
Microsatellite	27/67	21/86	30/93	138/88
UBC	29/38	29/41	16/69	79/72
Cytokeratin20	14/71	35/80	35/100	na
BTA	31/16	43/47	50/52	91/91
TPS	29/32	35/54	15/74	72/63
Cytology	239/17	274/34	201/58	861/95

## Data Availability

The datasets generated and/or analyzed during the current study are not publicly available as there are no public repositories for this type of dataset. The data are available from the corresponding author on reasonable request. The data supporting reported review was obtained from public data base which were found by searching google scholar or google.

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
