# Peer review of "Microsatellite Instability Analysis (MSA) for Bladder Cancer: Past History and Future Directions"

_ijms, 2021, doi:10.3390/ijms222312864_

Round 1
Reviewer 1 Report
This is an interesting review manuscript on microsatelite instability analysis (MSI) for baldder cancer detection. The authors have made a really good effort to describe the principle of MSI analysis and clinical impact for BlCa patients.
Major comments
- The references section and thus the text of the review article should be updated to include recent studies on the field. In fact, the authors have cited studies up to 2015 or 2016, while the most of them are published <2010. Moreover, the studies discussed on MSI clinical value for BlCa detection, see Table 1, are published <2010 and the vast majority of them published at ~2000. The authors have to carefully address this issue and update the reference section and studies presented.
- I believe that the section "4. Other assays for bladder cancer detection", including BTA, NMP22 and UroVision, do not add anything to the review. Moreover, these have been frequently presented in the related literature for BlCa detection. Thus, in a review article focusing on MSI, these tests, their discussion and Tables 2-5 do not help the readers.
- Instead of "4. Other assays for bladder cancer detection", the authors should discuss the prognostic and predictive value of MSI analysis for BlCa patients, and not only their value in BlCa detection.
- A figure describing the molecular basis of MSI (the spontaneous loss or gain of nucleotides from repetitive DNA tracts) will help the non experienced readers as well as the impact of the work.
Author Response
We very much appreciate review's insight and our response is provided below per each comment. The revised manuscript incorporates changes outlined in our response
- The references section and thus the text of the review article should be updated to include recent studies on the field. In fact, the authors have cited studies up to 2015 or 2016, while the most of them are published <2010. Moreover, the studies discussed on MSI clinical value for BlCa detection, see Table 1, are published <2010 and the vast majority of them published at ~2000. The authors have to carefully address this issue and update the reference section and studies presented.
In section 3.3, from 303-305, this issue has been address mentioning the fact that most of the studies were performed in early 2000’s.
In section 3.4, from 365-368, this issue has been address mentioning the fact that most of the studies were performed in early 2000’s and other important point in table.
- I believe that the section "4. Other assays for bladder cancer detection", including BTA, NMP22 and UroVision, do not add anything to the review. Moreover, these have been frequently presented in the related literature for BlCa detection. Thus, in a review article focusing on MSI, these tests, their discussion and Tables 2-5 do not help the readers.
Based on reviewer’s comments, section 4 and Table 4 and 5 are deleted with rearrangement of reference section. We have kept Table 2 and 3 as these two tables show important comparison between MSA and other bladder cancer biomarkers for its overall diagnostic accuracy and tumor grade dependent sensitivity.
- Instead of "4. Other assays for bladder cancer detection", the authors should discuss the prognostic and predictive value of MSI analysis for BlCa patients, and not only their value in BlCa detection.
We were mostly interested in MSA as a marker for future prognosis as related with predicting chance for recurrent disease. This has been discussed in section 3.6. as a separate subject.
- A figure describing the molecular basis of MSI (the spontaneous loss or gain of nucleotides from repetitive DNA tracts) will help the non experienced readers as well as the impact of the work.
New Figure 1 is added to the manuscript and new sentence explaining MSI with DNA replication error is added from line 106 to 121. Additionally, existing Fig 1 was changed into Fig 2: In line 149 and 508, Figure 1 was changed into Figure 2.
Reviewer 2 Report
Dear Authors,
current manuscript is an interesting narrative review of MSI capacity to detect urothelial cancer in Urine. Work is well conceived, but some minor points might be improved:
1) Introduction: from line 49 to line 80. Regarding urothelial cancer diagnosis and MSA there are no references at all. Cite at least guidelines
2) Methods: Your work is a good narrative review, and at least the criteria of search and selection of articles should be cited. If you want to analyze data as you did, putting together different studies it become a systematic review and meta-analysis of MSA techniques for detection and recurrence. Decide if you should keep it as a narrative review or to improve it in a systematic review and meta-analysis. In the last case, please provide all the necessary additional files and flowcharts and possibly provide a forest-plot
Author Response
We very much appreciate reviewer's excellent comments and we are providing our response per each of the comments as below. Also our response has been incorporated into attached revised manuscript
1) Introduction: from line 49 to line 80. Regarding urothelial cancer diagnosis and MSA there are no references at all. Cite at least guidelines.
Appropriate references have been added.
2) Methods: Your work is a good narrative review, and at least the criteria of search and selection of articles should be cited. If you want to analyze data as you did, putting together different studies it become a systematic review and meta-analysis of MSA techniques for detection and recurrence. Decide if you should keep it as a narrative review or to improve it in a systematic review and meta-analysis. In the last case, please provide all the necessary additional files and flowcharts and possibly provide a forest-plot
In introduction, we have clarified that this is a narrative review and also criteria for “selection of article” have been added in line (83-89)